# A Case Study of European Collaboration between the Veterinary and Human Field for the Development of RSV Vaccines

**DOI:** 10.3390/vaccines11071137

**Published:** 2023-06-23

**Authors:** Marga Janse, Giulia Sesa, Linda van de Burgwal

**Affiliations:** Athena Institute, VU Amsterdam, 1081 HV Amsterdam, The Netherlands; g.sesa@alumni.maastrichtuniversity.nl

**Keywords:** collaboration, KOLs, human and veterinary health, knowledge sharing, HRSV and BRSV, vaccine development

## Abstract

The One Health (OH) approach describes the interconnection between the health of animals, humans, and the environment. The need for collaboration between the veterinary and human fields is increasing due to the rise in several infectious diseases that cross human–animal barriers and need to be addressed jointly. However, such collaboration is not evident in practice, especially for non-zoonotic diseases. A qualitative research approach was used to explore the barriers and enablers influencing collaborative efforts on the development of vaccines for the non-zoonotic RSV virus. It was found that in the European context, most veterinary and human health professionals involved in RSV vaccine development see themselves as belonging to two distinct groups, indicating a lack of a common goal for collaboration. Next to this, the different conceptualizations of the OH approach, and the fact that RSV is not a zoonotic disease, strengthens the opinion that there is no shared need for collaboration. This paper adds insights on how, for a non-zoonotic situation, collaboration between human and veterinary professionals shaped the development of vaccines in both areas; thus, improving public health requires awareness, mutual appreciation, and shared goal setting.

## 1. Introduction

According to the One Health (OH) approach, the health of animals, humans, and the environment is intertwined, and therefore, various infectious diseases occurring across human–animal species barriers need to be addressed together. Clearly the need for collaboration between the veterinary and human fields is increasing due to the rise in human infectious diseases with a zoonotic origin, such as COVID-19, and the increase in antibiotic resistance threatening global public health [1]. The WHO, the European Commission’s Directorate-General for Health and Consumers, USA Centers for Disease Control & Prevention (CDC), and World bank have placed the OH approach high on their agendas [2]. However, in practice, collaboration between the human, animal, and environmental fields is only slowly increasing despite the creation of numerous networks of professionals in the fields of health and the environment (e.g., Med-Vet-Net, ArboZoonet, and NBIC) in order to bridge the communication gap between science and society [3].

One such case, where the OH approach is expected to be beneficial for collaboration, is the development of RSV vaccines. Respiratory syncytial viruses (RSV) in cattle (BRSV) and humans (HRSV) are two globally widespread, closely related, and highly infectious viruses [4,5,6,7]. Notably, the viruses are not zoonotic [8,9], which is commonly the main target for collaboration from the OH perspective [10]. In northern Europe, biannual cycles of RSV activity are known to lead to lower respiratory tract infections among infants, the elderly, and at-risk adults (e.g., those with immune system disorders, heart and lung diseases, and asthma) [11,12], and in the case of BRSV, among calves [13]. Across Europe, lower respiratory tract infections, caused by HRSV, are responsible for 42–45% of hospital admissions in children under two. Furthermore, HRSV pneumonia, which accounts for 6.7% of all newborn fatalities globally, is the second-largest cause of death in newborns under six months after malaria [14,15,16]. Similarly, BRSV is responsible for more than 60% of respiratory illnesses among beef and dairy herds with a mortality rate of BRSV up to 20% during severe outbreaks posing a global threat to the farming industry [17].

In contrast to delivering prophylactic medicines to all population groups, vaccines constitute a successful and cost-effective means of prevention but have not been successfully developed yet for all types of infectious diseases [18,19]. Hence, the availability of BRSV and HRSV vaccines is pivotal in addressing RSV infections. BRSV vaccines have been available for almost two decades but require substantial improvements given their administration mode and inability to induce long-term immunity [20], and despite there being several HRSV vaccination trials ongoing, there are currently no approved HRSV vaccines on the market. The RSV field, thus, seems to be an exemplary field for OH collaboration in the development of vaccines for both fields. Given the above, BRSV and HRSV pose a significant burden on global human and animal health. Based on their genetic relatedness, similar pathogenesis, comparable severity of the disease in the elderly, young children, and young calves, it is expected that much can be gained by collaboration, sharing knowledge and experiences between the human and veterinary fields, benefiting global health supported by the OH approach. This is in line with results from past collaboration successes for other infectious diseases, contributing to the development of the smallpox vaccine, canine distemper vaccine, and diphtheria antitoxin [3]. From these earlier successes it becomes clear that combating and controlling zoonoses and emerging infectious diseases requires an integrated approach and a common goal. In addition, the exchange of knowledge on disease treatment can lead to major savings in healthcare costs and to new scientific insights [2].

Results from our earlier research, using patents as indicators for collaboration, however, showed that there is no collaboration and cross-utilization of novel inventions, with no patents being co-owned by human and veterinary applicants [21]. Nevertheless, information regarding inventions described in former patents, for veterinary and human use, was found in a few successor patents, mostly describing techniques fit for both fields, showing that knowledge is being exchanged on a small scale. This patent search did not identify the causes, barriers, or opportunities for collaboration on RSV vaccine development.

Here, we address this knowledge gap by looking into the barriers and opportunities for collaboration among HRSV and BRSV professionals. Collaboration is defined as: ‘a mutually beneficial partnership between at least two actors (for example, individuals, organizations, and groups) working together to achieve a common goal or product’ [22]. To gain a better understanding of the collaboration between healthcare professionals working on RSV vaccines in the veterinary and human fields, we used the Common Ingroup Identity Model (CIIM). The CIIM describes a social-categorization-based perspective for improving group relationships and reducing bias by conceptualizing the relationship between individuals’ perceptions and behaviors toward groups, and the factors that influence them [23]. The core premise of the CIIM is that circumstances cause members, of various groups, to classify themselves as members of a single, more inclusive group, thus thinking in terms of our, we, or us. The concepts of the CIIM help to explore these circumstances. Sharing a common goal affects the degree of this group differentiation, and is influenced by the perception of this degree and the means of categorization [2]. The OH approach can be seen as a shared goal or common goal supporting professionals developing RSV vaccines, leading to more positive thoughts, feelings, and behaviors between individuals in the different groups. Collaborating on a common goal requires de-categorization, which means re-arranging existing groups, creating space to recognize the overlap with the other group, eventually leading to more collaboration. Following this reclassification, in which an overall group identity emerges, the formerly distinct groups will merge to form new subgroups. Members who see themselves as belonging to a larger, common whole will consciously classify themselves within that greater whole, reducing prejudice between groups, hence making collaboration a more ‘natural’ thing to do. Next to this, mutual differentiation is important for both groups to survive and operate separately, each playing a complementary and synergistic role within the framework of the common goal. Having a clear common interest also makes knowledge exchange more accepted because individuals can use each other’s expertise. Previous collaborative experiences will be supportive to having a good relationship with other groups and will increase attraction later. In contrast, when there is insufficient awareness of the potential benefits of collaboration (in our case, human and veterinary collaboration), healthcare professionals will place themselves in a certain category and identify with their ‘own’ profession. Lastly, as collaborations do not exist in a vacuum, external factors such as political will, geographical location, and the social and economic context also play a significant role [24]. This research, thus, aims to inform on improving collaboration for RSV-related unmet needs specifically, but also broader collaboration efforts for other non-zoonotic infectious diseases.

## 2. Materials and Methods

A qualitative research approach was used to explore barriers and enablers hampering or enhancing One Health collaboration for the development of RSV vaccines. This approach enabled us to identify and gain in-depth understanding of the views of both veterinary and human healthcare professionals concerning collaboration in practice.

### 2.1. Participant Recruitment

Renowned experts in both fields were invited to participate in an interview to gain their personal ideas [25]. Key Opinion Leaders (KOLs) were defined as experts working in one of the fields with different professional backgrounds, such as public health professionals, medical doctors, researchers, veterinarians, and those in the pharmaceutical industry. Such KOLs have direct experience with collaboration from practical experience on the one hand, and on the other hand, can convey the experiences of a broad network of stakeholders in this field. Initially, purposive sampling was performed by exploring several general RSV networks, such as the ReSViNET, identifying 28 potential participants. After this, LinkedIn networks were explored to recruit additional participants, resulting in six additional respondents. During the interviews, participants were asked to recommend names of other European experts or institutes working in the previously mentioned field. This sampling strategy led to a total of 45 potential participants.

### 2.2. Semi-Structured Interviews

Semi-structured interviews with open-ended questions were used for interviewing, allowing for in-depth exploration of topics that are potential mediators and impede collaboration between the veterinary and human sectors. The interview topics were based on the following concepts of the CIIM: shared aim/goal being the OH approach, collaboration awareness, mutual differentiations and de-categorization, group differentiation, knowledge sharing, past collaboration experiences [2]. Additionally, the concept of collaboration incentives was added as the broader context shaping the way collaboration takes place [24]. The interview questions were discussed and pilot tested with an expert from the veterinary RSV field before the actual interviews were conducted. Given the European setting of this study and COVID-19 situation at that moment (March–April 2022), all interviews were conducted using the online platform Microsoft Teams. The interviews primarily focused on filling in the gaps of knowledge on the practical experience with collaboration. Once the invitees had given consent, the interviews were conducted, and audio recorded using an iPhone. Field notes were made during and directly after the interviews. Participants were referred to as ‘#1, 2, 3 Vet or # 1, 2, 3 Hum, etc.,’ to maintain their anonymity.

### 2.3. Data Collection and Analysis

After the audio recording, all interviews were transcribed using the transcribing function of Word. The resulting text was double-checked and supplemented by manually transcribing, where required. Additionally, a summary of each interview was made to visualize the narrative behind individual interviews. Depending on the availability of the participants, a member check of interview summaries followed. The concepts from the CIIM provided the basis for creating categories and codes used for data analysis. Transcripts were read, the text was labeled with the identified codes. To avoid over-labeling, transcript fragments relevant to the study’s research objective but not fitting the pre-established categories, were highlighted and successively coded, generating 11 new codes. Overlapping codes (with similar connotation) were reduced, generating a final set of 80 codes. Ultimately, vertical, and horizontal coding was performed using Atlas.ti. to categorize the mentioned causes. The number of causes is depicted in Figure 1 and show that after interview 12, no new codes for causes emerged, meaning data saturation was reached.

## 3. Results

### 3.1. Participants

Out of 45 potential participants, 15 KOLs accepted the invitation to participate in this study. Five KOLs were specialists from the BRSV field, and 10 KOLs were specialists from the HRSV field. Data saturation of mentioned causes was reached after 12 interviews, as can be seen in Figure 1. The Y-axis represents the cumulative number of codes that emerged during data analysis.

Most participants (*n* = 11) were from Western Europe. A smaller number originated from Northern (*n* = 2) and Central Europe (*n* = 1). One of the participants currently worked in North America and was, despite the non-European professional setting, included in the research, given their understanding of the European context. No participants from Southern and Eastern Europe were able to join for an interview. The professional backgrounds of the KOLs can be found in Table 1.

### 3.2. Interviews

In total, 438 separate causes were mentioned concerning collaboration and knowledge sharing between the two fields. To categorize the causes, 80 codes were created, and these were divided into 26 groups. To further categorize the drivers and barriers for collaboration, five main themes were created following the CIIM: Group differentiation, Awareness, Common goal, OH approach, Collaboration.

### 3.3. Group Differentiation/Two Different Groups

When both groups of professionals see themselves as one group, according to the CIIM, sharing knowledge and collaboration will benefit [23]. Four Veterinary (Vet) and seven Human (Hum) KOLs, however, stated that, due to multiple reasons, professionals in both fields see themselves as two separate groups, mentioning the ‘us/them’ feeling as the main cause hampering collaboration.


*(#1Vet) ‘In my opinion human and veterinary healthcare professionals exist in two different universes.’*


Several causes were mentioned as generating the us/them feeling, conceivably leading to a lack of collaboration. First, one cause mentioned by four Hum KOLs is that it feels as if it is ‘detrimental’ even talking to the veterinary professionals mentioning ‘no urgency’, ‘no sense’, and ‘not useful’ as the cause. Furthermore, two other Hum KOLs indicated that the rapid advancements in the human field, specifically in the way of studying RSV and the HRSV vaccine development, has increased the motivation for human professionals to follow advances in the human field while lowering interest in the veterinary arena and the urgency to collaborate. In particular, collaborating with the veterinary sector and contributing to BRSV vaccine development is considered a diversion for the human pharmaceutical business, according to yet another Hum KOL, also mentioning that the human field lacks understanding of the advantages of learning from one another.


*(# 9Hum) ‘I think there have been so many developments in human studies, new ways of studying human diseases and so much more research activities concerning human disease, that it’s quite easy to just remain within our human bubble and to have relatively little contact with those who are working on veterinary aspects.’*


Second, the so-called self-interest was mentioned by three Vet KOLs and seven Hum KOLs referring to the limited awareness of what the veterinary field needs or has to offer because human professionals are looking to develop human vaccines only. Next to this, three other Vet and one Hum KOLs mentioned that human professionals feel more advanced, and according to one Vet and four Hum KOLs, both fields have separate interests with different events and organizations attracting the professionals.


*(#14Vet) ‘People in the human pharma think they are more ‘advanced’, as if they almost consider the use of knowledge or expertise from animal health as inferior’.*


Third, the status difference between veterinary and human health professionals leads to higher financial assets, and consequently, the human field has access to bigger, more expensive, and better laboratories supporting vaccine development research, as mentioned by three different Vet KOLs and two Hum KOLs. Having this higher group reputation, as mentioned by two Vet KOLS, the human field has more academic opportunities (e.g., publishing in journals with higher impact factors) and receives research funding faster. Operating in different work environments (separate universities, medical/clinics, separate pharma organizations) also increases the separation between the two fields according to two Vet KOLs. Fourth, according to one Vet and three Hum KOLs, human RSV professionals have a broader target population for HRSV vaccine development including infants, the adult population with medical conditions, the elderly, and pregnant women, increasing the ‘us’ versus ‘them’ feeling.

Adding to the above, human, and veterinary health professionals experience different pressures during vaccine development and disease prevention, as mentioned by one Vet KOL, given the varied weight attributed to a human versus a bovine death. Moreover, three Vet and three Hum KOLs mentioned that the two groups of professionals specifically differ in their perceptions of risk related to the development of vaccines. Safety in the human field of RSV vaccines is considered to be of vital importance, whereas the veterinary industry is less sensitive to risks.


*(#3Hum) ‘It’s acceptable that cows die, it is not acceptable that humans die (..)We tolerate risks better in calves in cattle than in humans.’*


Lastly the differences in the costs of vaccine development and the final product were mentioned by three Vet and three Hum KOLs.


*(#5Hum)*
*‘*
*Cost*
*of goods is very important for the veterinary field (..) For us (in the human field, red.) the*
*cost*
*of goods is not very important because normally the vaccine price can be $20 or $100 (..) For bovine vaccines it has to be very low, but I think nobody really ever did an accurate calculation.’*


### 3.4. Awareness

One factor that is crucial in enhancing collaboration according to the CIIM is ‘sufficient awareness’. This refers to whether one group perceives another as suitable for collaboration and the potential benefits of collaboration along with whether there is widespread knowledge of the current collaborations in the European context among stakeholders [23]. When asked specifically, two Vet and four Hum KOLs were aware of current collaborations between veterinary and human health institutes in Europe, as mentioned by two Vet and four Hum KOLs mostly entailing the use of animal models (calves) for testing human vaccine prototypes. According to two of these Hum KOLs, there is even collaboration between veterinary and human specialists on other disease-related aspects such as RSV pathophysiology.


*(#6 Hum) ‘It’s mainly the pathogenesis (we discuss with the veterinarians, red.) and what kind of vaccines have been developed and whether there is something that we could learn.’*


In contrast, three other Vet KOLs and eight Human KOLs mentioned that, in Europe, there is a ‘poor awareness of possible collaboration’ for the development of RSV vaccines in both fields.


*(#1Vet) ‘I don’t think that people run away for collaboration, but we (veterinary and human professionals, red.) live in different worlds.’*


### 3.5. Common Goal

According to the CIIM, sharing a common goal influences the degree of collaboration. One factor that positively influences the establishment of a common goal are positive earlier collaboration experiences [23]. Past collaboration experiences were mentioned by two Vet and three Hum KOLs referring to collaboration in the 1980s/90s focusing on partnerships in RSV research using the bovine model to test human vaccines and the development of BRSV vaccines, including with institutions on a European and global scale, namely the Pirbright Institute, GSK, NIH, INRAe, and Iowa State University. Yet the effects of such collaborations never led to lasting impacts, according to one of these Hum KOLs, because it was too difficult to develop safe vaccines for humans.


*(#9Hum) ‘The fundamental questions that we were addressing together in the 80/90s have essentially been addressed to a large extent (..) so now we have the tools and the programs that are more confined for the human species. It allows us to make a lot of progress without necessarily referring to the veterinary field’.*


Another factor that makes it challenging to set shared goals and collaborate for a ‘common good’, according to one Vet and one Hum KOL, is the lack of awareness surrounding the similarities of the BRSV and HRSV viruses. Additionally, two Vet and two Hum KOLs specifically mentioned that the differences in viruses are a reason to not collaborate.


*(#4Hum) ‘They (BRSV and HRSV, red.) are different viruses, there’s only about 70% sequence homology between bovine and human RSV. So, you’re targeting really two different viruses, although they are very similar.’*


Additionally, as mentioned by one Vet and three Hum KOLs, the changed public perception on performing animal research in the recent decade is hampering the formulation of a shared common goal for collaboration. One of these Hum KOLs mentioned that animal data are seen as misleading or impeding human vaccine research, lowering the threshold for moving to human studies.


*(#9Hum) ‘In the past there was an absolute rule that in order to go for clinical trials in humans you had to show data from at least two animal species (..) Over the last decade the perception is that sometimes data generated by doing research in animals can hold back or mislead human development (..) So there is a lower threshold for doing human studies. (..) and animal testing is more and more not done by now.’*


Although they did not mention a common goal, two Vet and four Hum KOLs did express willingness to work together in the RSV fields. As suggested by one Hum KOL, the Netherlands and the United Kingdom (UK) have several BRSV and HRSV experts concentrated in small areas, potentially transforming this willingness into action.


*(# 11Vet) ‘We could make bigger strides if we work more closely together.’*


This was supported by two Vet and two Hum KOLs stating that veterinary professionals are more open to, aware of, and supportive of the human field, and have a larger drive to collaborate. However, with the ultimate goal being slightly different for both groups (saving human lives from RSV or preventing disease in livestock), one KOL was also wary of the risk of failures in one domain in order to achieve the goal in the other.


*(#1Vet) ‘On (the) veterinary side, people are open. We collaborate for example with (organization name, red.) but are scared that if we have a problem in the bovine field this might impact the human side.’*


### 3.6. One Health Approach

The OH approach, when used as a common goal, can be expected to lead to greater collaboration [2]. However, although the existence of and familiarity with the OH approach was mentioned by five Vet and nine Hum KOLs, different focus points were mentioned on how collaborating, according to the OH approach, should be executed and where the focus should be put.

According to two of these Vet KOLs, the OH approach lacks a common definition, providing room for misinterpretations, and three Vet and two Hum KOLs emphasized that OH concerns animal-related conditions that directly influence human health, such as zoonotic diseases and antimicrobial resistance (AMR).


*(#7Hum) ‘I think you would find an OH Approach with antibiotics and antibiotic resistance. That’s clearly one of the drivers that make people work together for the common good’.*


Contrastingly, two Vet and three Hum KOLs indicated that veterinary and human health professionals should address health challenges, arising from the interconnection between animals, humans, plants, and the environment, together by adopting the OH approach. As this is not a widely shared meaning and RSV does not relate to the criterion of cross-species transmission, the OH approach is ‘not on the agenda’ for RSV according to one Vet and one Hum KOL

Moreover, European nations lack RSV-focused OH organizations, contributing to the lack of awareness of the potential of the OH approach for RSV. Different KOLs mentioned national OH centers focusing primarily on zoonotic diseases, reinforcing the previously stated misconceptions. Similarly, one Hum KOL mentioned that pharmaceutical companies also typically lack RSV-focused OH initiatives.


*(#11Vet) ‘My question would be: what do you think is a OH approach? National OH centers mainly focus on zoonosis, for example, and this (RSV) is not really a zoonosis.’*

*(#13Hum) ‘Because there are no immediate points of concern of cross-species transmission of RSV, there is also less urgency’.*


### 3.7. Collaboration

Mutual differentiation was not examined in depth due to the lack of a shared drive for collaboration as mentioned by the majority of most KOLs. Two Hum KOLs mentioned that two groups’ expertise merges in RSV vaccine research partnerships, concerning calve models, and that preconceptions make it difficult to achieve a mutual agreement. Categorization in groupings seems to depend on the domain of the professionals working in the human and veterinary health fields. Specifically, veterinary, and human health clinicians see themselves as two separate groups. Academics in both fields, on the other hand, feel part of a larger community according to one Vet and two Hum KOLs.


*(#13Hum) ‘There is a difference between veterinary and medical clinicians, and this is less different when it comes to academics in both fields.’*


Additionally, professionals in the veterinary field are considered too modest to acknowledge that they really can be of additional value for the human field, according to two different Vet and two Hum KOLs. This so-called ‘thinking in silos’, was mentioned by three Vet and seven Hum KOLs, and is specifically prevalent among human health professionals, according to one Vet and one Hum KOL.


*(#2Vet) ‘In the veterinarian field we are still too modest to acknowledge that we are really of additional value to the human field.’*


### 3.8. Knowledge Sharing

Despite the similarities between the BRSV and HRSV strains, as mentioned by three Vet and four Hum KOLs, comparable disease management mentioned by yet another Hum KOL, comparable vaccine development according to one Vet and one Hum KOL, and similar target groups, the effectuation of knowledge sharing depends on the attitude of ‘higher management’, according to one other Vet and three other Hum KOLs.


*(#5Hum) ‘I’m not sure how that is in other research organizations and in other industries, but it really depends on people at particular positions in higher management (when and what knowledge is shared with whom, red.).’*


Although, currently, few European collaborations are focused on using the bovine model to evaluate HRSV vaccine candidates, the value of the translation of the veterinary research findings into knowledge that can help the development of human vaccines was acknowledged by three Vet and eight Hum KOLs. Contrastingly, human professionals do not appear to be involved in veterinary RSV research involving the bovine model, despite having a wealth of information to be shared with the veterinary discipline. According to three Vet and seven Hum KOLs, this is because the human professionals see no long-term perspective and accordingly feel no need to create ways of knowledge exchange with the veterinary professionals. As a result, veterinarians’ knowledge benefits the human field rather than the other way around, focusing on ‘one-way’ communication, according to one of these Hum KOLs.


*(#4Hum) ‘Animal research helped me to design a safer or better RSV vaccine, I was not helping an animal researcher making a better or safer veterinary vaccine’.*

*(#12Vet) ‘There’s a lot of knowledge from the human field that could support the veterinary field, but I feel it’s unbalanced, the knowledge from the veterinary field is way more supporting the human field’.*


### 3.9. External Factors Hampering Collaboration

Collaborations do not exist in a vacuum, external factors such as political will, geographical location, and the social and economic context play a significant role [24]. According to five Vet and five Hum KOLs, bureaucratic, infrastructural, poor political attention, and budgetary constraints are affecting European collaboration. One of these Hum KOLs mentioned issues surrounding intellectual property protection, because the early disclosure of study results is an incentive for some parties. Next to this, as mentioned by one Vet and one Hum KOL, European regulatory authorities lack awareness of the value of the bovine model and have a belief that veterinary knowledge is inadequate.


*(#7Hum) ‘The bovine model is an excellent model to test vaccines, but the regulatory agencies are not used to seeing large animal models for efficacy studies (..) They always have relied on small animal models (..). So the pressure comes from the regulatory agencies not from the people who are doing the science’.*


With regards to the differences in the granting of financial grants in the different fields, two Vet and five Hum KOLs noted that conducting research using bovine models is very expensive, reducing the chance of funding for joint studies. In addition, as mentioned by one of these Hum KOLs, in Europe, only two locations offer the necessary infrastructure and staff to conduct bovine trials, which also contributes to the reluctance of human pharmaceutical firms to collaborate with veterinary professionals.

### 3.10. Boosting Collaboration

To resolve the barriers to collaboration, various incentives stimulating collaboration were mentioned by two Vet and one Hum KOLs, such as an economic boost through government projects and European projects, exposure to the other group at joint conferences and symposia, the creation of consortia, and joint training and education programs. Creating networking opportunities, clear expectations on knowledge exchange, strategic alliances, joint publications, and the formation of specialized teams were mentioned by two different Vet and three Hum KOLs as ‘favorable aspects’ contributing to knowledge sharing. Two different Vet KOLs and one Hum KOL mentioned setting expectations, establishing the ‘why’ to collaborate, for PhD students working in both fields as ‘favorable aspects’ that contribute to awareness for collaboration between the two fields.


*(# 1Vet) ‘The best way to make people collaborate is to have a source of funding for a research project where you need to have 50% of veterinary experts and 50% of human doctors or other human experts’.*


## 4. Discussion

Here we explore the reasons for limited collaboration between the human and veterinary fields in RSV vaccine innovation. We find that KOLs mentioned the lack of a shared common goal for collaboration between the human and veterinary fields, although in both domains, vaccines are developed for similar viruses. In the European context, KOLs from both veterinary and human health fields involved in RSV vaccine development see themselves as belonging to two distinct groups with little awareness of the ‘other’ group. Next to this, the different conceptualizations of the OH approach, and the fact that RSV is not a zoonotic disease strengthens the opinion that there is no shared need for collaboration.

In our earlier research, as a predictive marker for collaboration, we used a patent search to create an overview of shared inventions and IP for RSV vaccine development. Our results showed that there is little cross-use of human and veterinary technologies and shared patents, reflecting little collaboration between the two fields [21]. The patent search did not provide any insight into the motivations of the individual healthcare professionals in both fields. This study, therefore, addressed the more specific knowledge gap of the role and importance of individual elements for collaboration between the human and veterinary fields. By zooming in on RSV, as a specific subfield of non-zoonotic infectious diseases, we were able to better understand the specific causes and consequences of the limited collaboration and sharing of knowledge. Whereas previous research already indicated the importance of a common goal for collaboration between different groups, here we show that the distinction of human and veterinary professionals in two subgroups has even decreased the sense of contributing to the same goal [2].

Healthcare professionals emphasized their ‘social dilemma’ [26], as collaborating with the veterinary field would take away precious time from working on human RSV infections. Without a felt shared interest, collaboration thus fails to take place. In recent years, human healthcare has been even more influenced by developments such as specialization, the diffusion of high technology, tendencies of concentration and privatization, and increasing competitive pressure. This has led to a higher emphasis on economic considerations and the raising of ethical concerns, putting patients that are seeking healthcare in a vulnerable position. Physicians, therefore, face a specific responsibility to act for the health benefit of their patients [27]. Our findings confirm that the human field has taken a separate path for the development of vaccines for vulnerable humans, specifically young babies, and is speeding up its R&D, considering the benefits of collaboration with the veterinary professionals as not useful.

Collaboration between a wide diversity of stakeholders in general is a challenging aspect in the vaccine development process and even more challenging when it comes to working from an OH approach, encountering multiple challenges such as funding, policy, surveillance, defining, and execution of the OH approach [10]. Currently, the main focus for collaboration from the OH approach between the different areas, is on resolving the unmet needs related to zoonotic outbreaks that are becoming more common in human populations, leading to public health emergencies such as Ebola, avian influenza, Q-fever, and SARS [3,28]. Our results show that as RSV is not zoonotic, there is, specifically among human healthcare professionals, no urgency to collaborate from the perspective of emerging zoonotic diseases. The successes of earlier collaborations in understanding the similarities and differences between HRSV and BRSV provided the human field with tools and programs limited to humans, increasing their degree of independence, and decreasing the need to refer to the veterinary field. It is not incomprehensible that our results show that in addition, due to a lack of knowledge of the OH concept, there are misunderstandings about the interpretation of the OH approach as only concerning the solving of zoonotic outbreaks. These misunderstandings need to be resolved regarding the prevention of future pandemics, which requires the need for human and veterinary health professionals to work together. The OH approach requires a multisectoral, transdisciplinary and collaborative approach, and professionals from both fields have to participate in these discussions [29]. It will, thus, be of interest to conduct future research on how in-group considerations differ between human and veterinary professionals in the context of zoonotic diseases. Moreover, given the global nature of infectious diseases, another interesting lens could be the distinction between in-group considerations between actors from the Global North vs. the Global South.

Next to zoonoses, the OH approach contains more elements that need serious attention to achieve and maintain a balance between human and animal health in a healthy environment [1,30]. In both the human and veterinary fields, professionals have developed knowledge and experiences that are of value to the other field, and which can be used as a starting point and input for learning from each other. This mutual learning process will help facilitate the R&D that is still needed to resolve the unmet needs in both fields. For example, the human field is more advanced in the employment of virus-like particles. In contrast, the veterinary field has achieved more progress in vector vaccinology [21]. Both technologies can be employed in both contexts; thus, learning from each other could benefit both fields. Moreover, the use of calve models to test HRSV vaccines generates data that are of benefit for learning in both fields [31,32]. Our results show that the veterinary field has a lot to offer and is willing to support the human field with the numerous novel technologies they have already employed [33].

Although veterinary medicine is at the forefront of viral vectored vaccinology, bringing these cost-effective vector-based vaccines that induce robust long-lasting immune responses to the veterinary market is critical to advancing the technology in favor of public health, while replicating the success to the human settings is still challenging due to safety concerns [34].

Increasing collaboration requires time, and to improve collaboration, it is critical to enhance veterinary professionals’ exposure to medical professionals and vice versa, as experts must first get to know one another before collaborating. Education in the OH approach must become a priority, thus breaking down the conventional disciplinary silos of human medicine, veterinary medicine, environmental health, public health, and the social sciences [10,35,36]. Younger generations may play an essential role in bridging these disciplinary silos. Current human-oriented conferences must include more veterinary issues and specialists, given human health professionals’ lack of appreciation of the benefits of collaboration. However, solely organizing conferences is insufficient; for attending such events, clear shared goals are crucial [37]. Another way of stimulating collaboration is governmental financing that is restricted to research programs involving both HRSV and BRSV and entailing an equal number of veterinary and human health professionals.

Government funding is limited, which is a factor potentially increasing competition rather than cooperation among the human and veterinary fields [38], so it is pivotal to raise the urgency and political focus around RSV, for example, by stressing the burden of HRSV and BRSV on global health and farming, as well as the advantages of prevention. European initiatives and grants have already been used to foster collaborations among different institutes to develop vaccines. For example, the SAPHIR (European Commission, 2020) project financed by Horizon 2020, focused on BRSV vaccine research, among other animal conditions. Other examples where opportunities led to the reforming of the political agenda, were the Salmonella and Campylobacter disease outbreaks in Sweden, leading to the development and implementation of new legislation [39].

Finally, establishing a shared meaning on the OH approach requires facilitating a flow of information between the human and veterinary sectors on similar health issues. In this regard, it is essential to involve OH organizations, because a common goal is not established until it is concrete, activating the mechanisms necessary to create an overarching identity and create mutual benefits [40].

## 5. Conclusions

We conclude that collaborating to innovate and develop new RSV vaccines, even for the common good, is not as logical as it might seem given the starting point of this research, which was the assumption that the development of vaccines for similar viruses such as BRSV/HRSV, causing significant health problems in both fields, would lead to collaboration.

By interviewing 15 KOLs working in a European setting to obtain a greater insight into their thoughts and feelings on collaboration, it became clear that the presumed collaboration in the end was not taking place for clear reasons. Overall, when it comes to developing vaccines in both areas, and thus improving public health as a whole, more work on awareness, appreciation, and shared goal setting is needed to achieve this.

## Figures and Tables

**Figure 1 vaccines-11-01137-f001:**
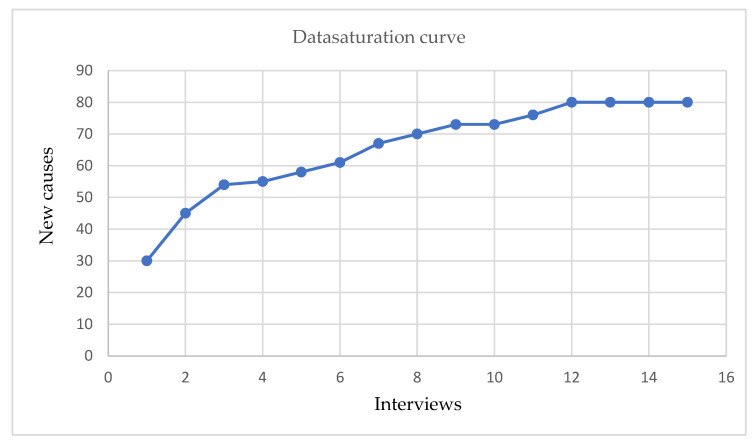
Data saturation was reached after 12 interviews.

**Table 1 vaccines-11-01137-t001:** Professional Backgrounds of five BRSV and 10 HRSV KOLs.

Interviewee	BRSV or HRSV KOL	Professional Background and Geographical Location
# 1	Veterinary health	Academia/Northern Europe
# 2	Veterinary health	Academia/Western Europe
# 3	Human Health	Medical Doctor/Clinic/Western Europe
# 4	Human Health	Medical Doctor/Clinic/Western Europe
# 5	Human Health	Pharma/Industry/Western Europe
# 6	Human Health	Public health/Government/Northern Europe
# 7	Human Health	Pharma/Industry/Western Europe
# 8	Human Health	Public health/Government/ Western Europe
# 9	Human Health	Medical Doctor/Clinic/Western Europe
# 10	Human Health	Medical Doctor/Clinic/Western Europe
# 11	Veterinary health	Researcher/Academia/Western Europe
# 12	Veterinary health	Pharma/Industry/Western Europe
# 13	Human Health	Researcher/Academia/Central Europe
# 14	Veterinary health	Pharma/Industry/Western Europe
# 15	Human Health	Pharma/Industry/North America

## Data Availability

Not applicable.

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
