# Peer review of "A Case Study of European Collaboration between the Veterinary and Human Field for the Development of RSV Vaccines"

_vaccines, 2023, doi:10.3390/vaccines11071137_

Round 1

Reviewer 1 Report

1. Results: The paper can be greatly improved by including some of the data from statistical analysis. Example, those supporting the first conclusion.

2. It explores an important link between human and veterinary collaboration however, it does not expound on the importance of HRSV and BRSV.

3. It explores the possibility and implication of one health approach in regards to HRSV and BRSV collaboration in vaccine development.

4. Add statistical analysis in;

i.  relation to vet KOLs and Hum KOLs

ii. order to strongly support the first conclusion

Further controls should be considered:

look into statistical analysis of the new causes shown in Fig 1., and their importance / contribution in the subject matter. 

5. While the author has provided strong arguments the evidence presented is shallow. This can be improved by presenting the statistical analyses conducted.

6. Additional comments on the tables and figures:

Figure #1 cannot be related to any of the methods. 

Reviewer 2 Report

In the manuscript titled” A case study of European collaboration between the veterinary and human field for the development of RSV-vaccines”, author adopts a qualitative research approach to explore barriers and enablers influencing collaborating on the development of RSV vaccines and found that there is no shared common goal for collaboration between the human and veterinary field. When it comes to developing vaccines in both areas and thus improving public health as a whole more work on awareness, appreciation, and shared goal-setting is needed to achieve this. This manuscript has certain practical significance. However, there are many problems in the manuscript, which need further revision and improvement. The specific amendments are as follows:

1. Briefly explain the significance of this manuscript in the abstract.

2. Line 29, why is global in parentheses, is it emphasized or has some special meaning, if not, please remove it.

3. Language issues deserve attention, for example, line 96 ”former separate groups form for example a new 96 subgroup” and lines 108-109 ”the social and economic context play also a significant role.”

4. The italics in the last paragraph of the introduction are unnecessary.

5. Please change the title of the manuscript under Materials and Methods to Palatino Linotype font, keep the font color black, and remove the underline.

6. Please change "6" to "six" in line 123. There are still many such mistakes in the manuscript. Please check them carefully and correct them.

7. Covid-19 or COVID-19, please unify the writing style, line 28 and line 128.

8. Line 141 ,#1,2,3Vet or # 1,2,3Hum 2, 3 etc.or #1, 2, 3 Vet or # 1, 2, 3 Hum 1, 2, 3 etc.” Make sure you write it correctly.

9. Please change the font in line 154.

10. Please correct the error in line 160.

11. The top of line 170 is presented as a table and not a figure, and the table should be strictly in the standard form of a three-line table.

12. Please revise the structure of materials and methods in the manuscript, such as the title of interview results should not appear under the results. There are many problems with the formatting below the Interview Results, such as unnecessary headings 1), 2), 3), 4), 5), avoiding blue font, multiple spaces, and language issues.

13. What is the abbreviation of AMR in line 306?

14. Only the first occurrence of the word should be marked with the full spelling and abbreviation, and the subsequent occurrences are presented in abbreviated form. For example, in line 494 "OH" and line 496 "One Health", check the manuscript carefully for similar problems.

Moderate editing of English language required

Reviewer 3 Report

It was a pleasure to review this very insightful paper - showing current behaviour of a range of KOLs. The non-zoonotic choice - with diseases being significant in both sectors - was a good one to gauge collaboration in relation to technological approach in developing vaccines - and exploring beyond.

It would be interesting to conduct work using CIIM since more collaboration would be expected with a zoonosis example. Also potential application in comparing between global north and global south contexts

I can only find one other example Eussen et al BMC-Vet.Res.2017 using the CIIM. there could be more application in OneHealth

The only query I have is related to age of KOL - assumed (appropriately!) biased to older as more experienced but wondering if younger KOLs had different opinion within participants - The young & upcoming may change this dynamic since OneHealth and different teaching methods may positively influence OH/collaboration.

Few minor notes on text

L31 US - Centers for Disease Control &  Prevention is CDC (cdc.gov)

L160 'error…'
